# Cycloguanil and Analogues Potently Target DHFR in Cancer Cells to Elicit Anti-Cancer Activity

**DOI:** 10.3390/metabo13020151

**Published:** 2023-01-19

**Authors:** Jennifer I. Brown, Peng Wang, Alan Y. L. Wong, Boryana Petrova, Rosanne Persaud, Sepideh Soukhtehzari, Melanie Lopez McDonald, Danielle Hanke, Josephine Christensen, Petar Iliev, Weiyuan Wang, Daniel K. Everton, Karla C. Williams, David A. Frank, Naama Kanarek, Brent D. G. Page

**Affiliations:** 1Faculty of Pharmaceutical Sciences, University of British Columbia, Vancouver, BC V6T 1Z3, Canada; 2Department of Pathology, Boston Children’s Hospital, Boston, MA 02115, USA; 3Harvard Medical School, Boston, MA 02115, USA; 4Harvard/MIT MD-PhD Program, Harvard Medical School, Boston, MA 02115, USA; 5Department of Hematology and Medical Oncology, Emory University, Atlanta, GA 30322, USA; 6The Broad Institute of Harvard and MIT, Cambridge, MA 02142, USA

**Keywords:** dihydrofolate reductase, folate metabolism, target engagement, cancer therapy, cycloguanil, breast cancer, STAT3

## Abstract

Dihydrofolate reductase (DHFR) is an established anti-cancer drug target whose inhibition disrupts folate metabolism and STAT3-dependent gene expression. Cycloguanil was proposed as a DHFR inhibitor in the 1950s and is the active metabolite of clinically approved *plasmodium* DHFR inhibitor Proguanil. The Cycloguanil scaffold was explored to generate potential cancer therapies in the 1970s. Herein, current computational and chemical biology techniques were employed to re-investigate the anti-cancer activity of Cycloguanil and related compounds. In silico modeling was employed to identify promising Cycloguanil analogues from NCI databases, which were cross-referenced with NCI-60 Human Tumor Cell Line Screening data. Using target engagement assays, it was found that these compounds engage DHFR in cells at sub-nanomolar concentrations; however, growth impairments were not observed until higher concentrations. Folinic acid treatment rescues the viability impairments induced by some, but not all, Cycloguanil analogues, suggesting these compounds may have additional targets. Cycloguanil and its most promising analogue, NSC127159, induced similar metabolite profiles compared to established DHFR inhibitors Methotrexate and Pyrimethamine while also blocking downstream signaling, including STAT3 transcriptional activity. These data confirm that Cycloguanil and its analogues are potent inhibitors of human DHFR, and their anti-cancer activity may be worth further investigation.

## 1. Introduction

Disrupting folate metabolism is one of the oldest and most effective strategies to treat cancer. The dihydrofolate reductase (DHFR) inhibitor methotrexate (MTX) was one of the first chemotherapies to be discovered and has been consistently used to treat various types of cancer since the late 1940s [1]. Classic DHFR inhibitors, including MTX, Pemetrexed, and Pralatrexate, mimic dihydrofolate (DHF) and occupy the DHF binding site of DHFR (Appendix A) [2]. 

While DHFR is one of the ‘oldest’ anti-cancer drug targets, more recent research has illuminated new inhibitors and polypharmacological effects that offer a more thorough understanding of DHFR biology and inhibitors. For example, the use of target engagement technologies such as Thermal Proteome Profiling (TPP)) helped confirm thymidylate synthase as an additional target for MTX [3]. Additionally, similar techniques helped demonstrate that the anti-malaria drug Pyrimethamine (Pyr) selectively inhibits human DHFR in cancer cell lines, which is linked to its downstream inhibition of STAT3 signaling and underlies its promising anti-cancer activity [4,5,6]. DHFR and its high-affinity inhibitors have been routinely used as positive controls when developing target engagement technologies [7,8,9,10,11]. This, in turn, has provided an ideal tool-kit for studying DHFR inhibitors and has prompted the discovery of new DHFR-focused target engagement assays, such as measuring DHFR accumulation following inhibitor treatment as an indicator of target engagement [4]. It is within this fruitful landscape that previously identified DHFR inhibitors were re-investigated for their ability to inhibit human DHFR in biochemical assays and engage DHFR in cancer cell lines.

Cycloguanil (Cyc) is the active form of the anti-malaria drug Proguanil which targets plasmodial DHFR to prevent parasitic infection. Activation of Proguanil to Cyc requires oxidation in the liver by cytochrome P450 enzymes [12]. Variability in cytochrome p450 expression in different patient populations contributes to the variable prophylactic efficacy of Proguanil [12]. The chemical structures of Cyc and Pyr are highly similar, with both compounds possessing a diamino-, nitrogen-containing heterocycle appended to a chlorophenyl substituent (Figure 1a). Cyc was originally identified as an inhibitor of human DHFR in the 1970s when it and many Cyc analogues were explored for their DHFR inhibitory activity and their ability to prolong life in rat models of leukemia [13,14,15]. Despite showing promising activity in these early experiments, these compounds arise in relatively few citations, suggesting a lack of further development, summarized in Appendix A. Thus, using modern in silico methods, archived screening data, and state-of-the-art chemical biology techniques, Cyc and its known triazine-containing analogues were herein revisited for their inhibitory activity against human DHFR and as potential cancer therapies. While similar potency was observed in biochemical experiments, potent target engagement was observed in cell-based assays suggesting that these triazine compounds may still be promising DHFR inhibitors with interesting biological and anti-cancer properties.

## 2. Materials and Methods

### 2.1. Cell Culture

Human epithelial breast adenocarcinoma cell lines MDA-MB-468 (HTB-132), MCF-7 (HTB-22), and MDA-MB-231 (HTB-26) were obtained from American Type Culture Collection (ATCC). Cells were routinely grown in Dulbecco’s Modified Eagle’s Medium (DMEM, Corning) supplemented with 4.5 g/L glucose, L-glutamine, sodium pyruvate, and 10% fetal bovine serum (FBS, Fisherbrand) at 37 °C with 5% CO_2_ in a humidified incubator.

### 2.2. Compounds

All compounds were prepared in 10 mM DMSO. Pyrimethamine (D897, AK Scientific, Union City, CA, USA), methotrexate (J10045, AK Scientific, Union City, CA, USA), and cycloguanil (7651AA, AK Scientific, Union City, CA, USA) were commercially purchased. Most cycloguanil analogues were received from the National Cancer Institute (NCI) Developmental Therapeutics program [16]: 3062 (etoprine), 123032, 127159, 127153, 128184, and 139105 (triazinate). Analogue 3077 was synthesized according to previously published protocols, as described in Appendix A. Compounds were locally indexed and visualized using Instant JChem (22.6.0) (ChemAxon).

### 2.3. Docking

Compounds were docked into the folate binding pocket of the DHFR crystal structure PDB 1U72 using GLIDE (Maestro, Schrödinger) [17]. Cyc analogues were downloaded from the National Cancer Institutes (NCI)—Developmental Therapeutics Program chemical compound dataset (2016 release) [16]. Triazine-containing structures were identified using the structure search function (Instant JChem) and were extracted for docking studies. Structures were prepared for docking using LigPrep. The DHFR co-crystal structure contained both nicotinamide adenine dinucleotide phosphate (NAPDH, which was removed for docking) and MTX. The protein crystal was prepared for docking using standard workflows. The co-crystallized MTX was used to set the center of a 10 × 10 × 10-angstrom docking grid. As a positive control, MTX was also subjected to docking experiments and compared to the co-crystalized ligand through the determination of the RMSD using the DockRMSD tool (v1.1) [18].

### 2.4. Expression and Purification of Human DHFR

Expression and purification of human 6 × His-DHFR was performed by the Protein Science Facility of Karolinska Institutet (Department of Medical Biochemistry and Biophysics, Biomedicum, Stockholm, Sweden). Briefly, the gene for human DHFR was produced by GeneArt (Thermofisher) with an N-terminal 6 × His-tag and cleavable tobacco etch virus (TEV) site and was cloned into the pNIC-Bsa4 expression plasmid. The construct was transformed into BL21 (DE3) T1R pRARE2, and expression was induced at OD 3 with 0.5 mM isopropyl ß-D-1-thiogalactopyranoside (IPTG) and grown 16 h at 18 °C. Cells were harvested by centrifugation at 4500× *g* and suspended in lysis buffer (100 mM HEPES, 500 mM NaCl, 10 mM imidazole, 10% glycerol, 0.5 mM tris(2-carboxyethyl)phosphine (TCEP), pH 8.0) supplemented with EDTA-free protease inhibitor cocktail (PIC, Roche) and benzonase nuclease cell suspension (50 μL/mL) at 1.5 mL buffer per gram wet cell pellet. Resuspended cells were frozen at −80 °C to aid in lysis. The cell suspension was thawed in a room temperature water bath and sonicated on ice in 4-s intervals at 80% amplitude for 4 min. Cellular debris and insoluble components were separated by centrifugation at 49,000× *g* for 20 min. The soluble fraction was removed and clarified by filtration (0.45 μm).

Clarified lysate was loaded onto a 5 mL HisTrap HP (GE Healthcare) that was pre-equilibrated with wash buffer 1 (20 mM HEPES, 500 mM NaCl, 10 mM imidazole, 10% glycerol, 0.5 mM TCEP, pH 7.5). Subsequent washes with wash buffer 1 and wash buffer 2 (wash buffer 1 with 50 mM imidazole) were performed. Bound protein was eluted (20 mM HEPES, 500 mM NaCl, 500 mM imidazole, 10% glycerol, 0.5 mM TCEP, pH 7.5) and subsequently loaded onto a HiLoad 16/60 Superdex 75 (GE Healthcare) size exclusion column pre-equilibrated with gel filtration buffer (20 mM HEPES, 300 mM NaCl, 10% glycerol, 0.5 mM TCEP, pH 7.5) to remove imidazole. The resulting purified protein was concentrated using a Vivaspin concentration filter with a 10 kDa molecular weight cut-off. The concentration of purified protein was determined by measuring the A_280_ using a Nanodrop and using the extinction coefficient 26,930 M^−1^ cm^−1^. The final fraction was determined to be >90% pure, as monitored by SDS-PAGE.

### 2.5. DHFR Enzymatic Inhibition Assay

Reactions containing 200 nM purified human DHFR, 137.5 μM dihyrdrofolate (DHF) substrate, and varying concentrations of inhibitor (0.091–200 μM) were prepared in DHFR buffer (50 mM Tris, 50 mM NaCl, pH 7.4). Working stocks of each inhibitor were created so that the final concentration of DMSO in each reaction was 2% in a final reaction volume of 100 μL. A DMSO control reaction containing 200 nM DHFR, 137.5 μM DHF, 2% DMSO, 125 μM NADPH, and no inhibitor was performed. A no-DHFR control reaction containing 2% DMSO, 137.5 μM DHF, and 125 μM NADPH was also performed. Reactions were prepared on ice in 384-well plates. To initiate the reaction, the NADPH cofactor (125 μM) was added. Consumption of NADPH by measuring absorbance at 340 nm was used to monitor the progress of the reaction. Readings were taken every 5 min for 1 h at 37 °C using a Synergy MX microplate reader (BioTek).

Percent inhibition relative to the no-inhibitor control was determined using Equation (1).
(1)$$\%\;Inhibition=\frac{A_{340}Inhibitor-A_{340}DMSO\;Control}{A_{340}No\;DHFR-A_{340}DMSO\;Control}\times 100$$

Percent inhibition data were then fit to the four-parameter logistic model (Equation (2)) in GraphPad Prism 9.4.1 to determine half-maximal inhibitory concentrations (IC_50_):(2)$$y=\frac{Bottom+\left(Top-Bottom\right)}{{\left(1+\frac{IC_{50}}{x}\right)}^{Hill\;slope}}$$
where *y* represents the percent inhibition, *x* represents the concentration of inhibitor (μM), the bottom and top represent the bottom and top plateaus, and the Hill slope represents the slope factor. Each experiment contained duplicates of each sample, and each experiment was performed 2 times (*n* = 2). An ordinary one-way ANOVA was used to test for statistical significance between treatments.

### 2.6. NCI-60 Sensitivity Correlation Analyses

Growth inhibition of 50% (GI_50_) data were accessed from the NCI-60 Human Tumor Cell Lines Screen database (https://dtp.cancer.gov/public_compare/ (accessed on 28 October 2022)) [19]. Datasets were pruned and points removed where GI_50_ values were equal to the maximum tested concentration. To analyze sensitivity profiles for compounds, GI_50_ values were compared using a correlation matrix in GraphPadPrism 9.4.1 to give Pearson Correlation Coefficients (r-values). Statistical significance is reported using a two-tailed statistical analysis to generate *p* values. Similarly, individual compound GI_50_ values were compared to MTX across different cancer cell types by performing correlation analysis to generate r-values using two-tailed statistical analysis to generate *p*-values. A plot of breast cancer GI_50_ values was prepared in which GI_50_ values that were equal to the highest concentration tested were omitted from the graph. These instances are indicated below the labels on the x-axis.

### 2.7. Viability Assays

The incorporation and metabolism of resazurin to resorufin was used to monitor cell viability [20]. MDA-MB-468, MCF7, and MDA-MB-231 cells were grown in 96-well plates with a total of 5000 cells per well in a total volume of 180 μL for 24 h, as described above. After 24 h of growth, cells were treated with increasing concentrations of compounds (0.0457–100 μM) for 72 h. A vehicle control was also performed. All treatments had a final concentration of 1% DMSO. The volume of treatment added to each well was 20 μL. For folinic acid rescue experiments, 5000 cells were treated with 10 μM compound with or without 300 μg/mL folinic acid (Supelco). After 72 h of treatment, resazurin was added to a final concentration of 44 μM, and cells were incubated for an additional 4 h at 37 °C. The fluorescence of resorufin was monitored on a Synergy MX microplate reader (BioTek) using an excitation wavelength of 540 nm and an emission wavelength of 600 nm. Percent viability relative to the DMSO control was determined. Each treatment was performed in triplicate, and each individual experiment was repeated three times (*n* = 3).

### 2.8. Wound Healing Assay

MDA-MB-231 cells were seeded in a 96-well Essen Image Lock plate (Essen Bioscience) at a density of 60,000 cells/well and cultured in DMEM medium supplemented with 10% Fetal Bovine Serum until 90% confluence at 37 °C and 5% CO_2_.

Confluent monolayers were scratched by using a 96-pin WoundMaker™ (BioScience Inc, Ann Arbor, MI, USA). Cells were then washed with DMEM three times to remove detached cells and debris. Compounds were diluted in DMEM medium and were added to cells at a final concentration of 10 µM for 24 h along with blank medium including 1% DMSO. Wound images were acquired by the IncuCyte™ software system (Essen BioSciences) every 3 h. Data were processed and analyzed using IncuCyte™ Scratch Wound Analysis Software Module. Data are presented as the Relative Wound Density, which represents the ratio of the cell density at the wound area relative to the cell density outside of the wound area over time. An ordinary one-way ANOVA with a Dunnett multiple comparison test was performed to detect significant differences between each treatment and the DMSO control. Each experimental condition was evaluated in quadruplicate in three independent assays (*n* = 3).

### 2.9. Lysate Cellular Thermal Shift Assay (CETSA)

Trypsinized MDA-MB-468 cells were harvested by centrifugation at 200× *g* for 5 min and washed with PBS supplemented with EDTA-free PIC (Roche) and flash frozen for storage at −80 °C. Cells were lysed by resuspension in 60 μL PBS/PIC per 1 × 10^6^ cells. Cells were lysed by three freeze-thaw cycles, where cells were frozen in an ethanol/dry ice bath for 3 min, immediately followed by incubation in a 37 °C water bath for 3 min. The insoluble fraction was removed by centrifugation at 20,000× *g* for 20 min at 4 °C. The supernatant containing the soluble lysate was removed and saved for subsequent steps.

To determine the melting temperature of DHFR, reactions containing 55 μg MDA-MB-468 lysate and 1% DMSO in 28 μL total DHFR buffer were incubated in a gradient from 37–81 °C for 3 min in PCR tubes in a DNA Engine Dyad Peltier Thermal Cycler (BioRad). After incubation, each reaction was transferred to a 1.5 mL tube and centrifuged at 20,000× *g* for 20 min at 4 °C to remove the insoluble proteins. After centrifugation, 18 μL of supernatant was carefully removed from each reaction, without disturbing any pelleted aggregates, and added to 4.5 μL of 5× SDS-PAGE loading buffer. 11 μL of each reaction was loaded onto a gel in duplicate, and proteins were detected by Western blot (as described below). Resulting band densities were fit to the Boltzmann equation to determine the melting temperature (*T*_m_):(3)$$y=Bottom+\frac{\left(Top-Bottom\right)}{1+e^{\left(\frac{T_{m}-x}{Slope}\right)}}$$

To rescue the thermal degradation of DHFR by inhibitors, samples containing 10 μM inhibitor, 55 μg lysate, and 1% DMSO in a total volume of 28 μL in DHFR buffer were prepared in PCR tubes and incubated at room temperature for 20 min. Two DMSO-only controls were also prepared. Reactions containing inhibitors were then incubated at 45 °C for 3 min. One DMSO-only control was also heated at 45 °C for 3 min, and the other control was heated at 37 °C for 3 min. The soluble protein fraction was separated from the insoluble proteins by centrifugation, and proteins were separated by SDS-PAGE, as described above, and analyzed via Western blot, as described below. Each experiment was repeated 3 times.

### 2.10. DHFR Accumulation Assay

Trypsinized MDA-MB-468 cells were seeded into 6-well plates at a density of 300,000 cells per 2 mL of media and allowed to adhere for 24 h. Cells were then treated with the indicated inhibitor at concentrations from 0.001 μM to 10 μM or from 0.012 nM to 1 nM. The final concentration of DMSO in each well was 1%. A DMSO-only control was also performed on each individual plate. Cells were then trypsinized, harvested by centrifugation at 200× *g* for 5 min at 4 °C, and washed once with PBS/PIC. Cell pellets were flash frozen using an ethanol/dry ice bath and stored at −80 °C prior to lysis. Each cell pellet was lysed as described above using 50 μL PBS/PIC per cell pellet. Samples were then analyzed by Western blot, as described below, using 30 μg lysate per lane. Band intensities of DHFR were normalized to the loading control SOD1 via densitometry using ImageJ (Fiji), and each DHFR band was normalized to the DMSO control. Fold-change in DHFR band intensity was plotted against inhibitor concentration (μM). Each experiment was performed two separate times as individual biological replicates (*n* = 2).

### 2.11. Western Blot

Proteins were electrophoretically separated on a 10% gel, and proteins were transferred to a methanol-activated PVDF membrane using the Trans-Blot Turbo transfer system (Bio-Rad) for 7 min at 1.3 A and 25 V in Trans-Blot Turbo transfer buffer (Bio-Rad). Membranes were blocked with 5% skim milk *w/v* in TBS-T (20 mM Tris, 150 mM NaCl, 0.05% Tween-20) for 1 h at room temperature. Primary antibodies were added for 1 h at room temperature in TBS-T using the following concentrations: anti-DHFR (sc-377091, Santa Cruz; 1:200); anti-SOD1 (sc-17767, Santa Cruz; 1:3000); anti-thymidine phosphorylase (TP) (sc-47702, Santa Cruz; 1:100). The horseradish peroxidase (HRP)-linked anti-mouse IgG secondary antibody (NA931, Cytiva; 1:3000) was added for 1 h at room temperature in 5% skim milk *w/v* in TBS-T. Signal was detected by adding Immobilon Forte Western HRP Substrate (Millipore), and chemiluminescence was captured on a Sapphire Biomolecular Imager (Azure Biosystems). Band densities were determined using ImageJ (Fiji) [21]. The band intensity of DHFR was normalized to the thermostable loading control SOD1 [22].

### 2.12. Liquid Chromatography/Mass Spectrometry (LC/MS)-Based Metabolite Profiling

#### 2.12.1. Sample Preparation for LC/MS Analysis of Polar Metabolites from MDA-MB-231 Celles

For characterization by mass spectrometry, MDA-MB-231 cells were seeded at a density of 3 × 10^5^ cells per well in 6-well plates. After 24 h of growth, cells were treated with either 10 μM inhibitor (MTX, Pyr, Cyc, or 127159) or 10 μM inhibitor plus 300 μg/mL folinic acid for 24 h. DMSO only and DMSO + folinic acid controls were also performed. Per condition, biological quadruplicates of approximately one million cells were harvested, washed briefly with 1 mL of 0.9% NaCl (prepared in LC/MS-grade water Fisher Scientific W6500), and extracted in 500 μL prechilled extraction buffer (80% LC/MS-grade methanol, 20% 125 mM ammonium acetate, 12.5 mM sodium ascorbate prepared in LC/MS-grade water and supplemented with aminopterin (catalog no.: 16.330; Schircks Laboratories) and isotopically labeled internal standards (17 amino acids and reduced glutathione [Cambridge Isotope Laboratories; MSK- A2-1.2 and CNLM-6245-10])) using a cell scraper. After centrifugation for 10 min, 4 °C, at maximum speed on a benchtop centrifuge (Eppendorf), the cleared supernatant was transferred to a new tube and dried using a nitrogen dryer (Reacti-Vap™ Evaporator, Thermo Fisher Scientific, TS-18826) while on ice. Once the drying process was completed, samples were reconstituted in 50 µL QReSS water (Cambridge Isotope Laboratories, MSK-QRESS-KIT) by brief sonication in a 4 °C water bath. Extracted metabolites were spun for 3 min, 4 °C, at maximum speed on a bench-top centrifuge, and cleared supernatant was transferred to LC/MS microvials (National Scientific, C5000-45B). Three microliters of each sample were pooled and serially diluted 3- and 10-fold to be used as quality controls throughout the run of each batch. Unlabeled QReSS metabolites (Cambridge Isotope Laboratories, MSK-QReSS-US-KIT) and unlabeled amino acid mix standard (Cambridge Isotope Laboratories, MSK-A2-US-1.2) were also included in the run to ensure high-quality metabolomics results.

#### 2.12.2. Chromatographic Conditions for LC/MS

One microliter (equivalent to 2 × 10^4^ cells) of the reconstituted sample was injected into a ZIC-pHILIC 150 × 2.1 mm (5 µm particle size) column (EMD Millipore) operated on a Vanquish™ Flex UHPLC Systems (Thermo Fisher Scientific, San Jose, CA, USA). Chromatographic separation was achieved using the following conditions: buffer A was acetonitrile; buffer B was 20 mM ammonium carbonate, 0.1% ammonium hydroxide. Gradient conditions were: linear gradient from 20% to 80% B; 20–20.5 min: from 80% to 20% B; 20.5–28 min: hold at 20% B. The column oven and autosampler tray were held at 25 °C and 4 °C, respectively.

#### 2.12.3. Orbitrap Conditions for Targeted Analysis of Polar Metabolites

Mass spectrometry data acquisition was performed using a QExactive benchtop orbitrap mass spectrometer equipped with an Ion Max source and a HESI II probe (Thermo Fisher Scientific, San Jose, CA, USA) and was performed in positive and negative ionization mode in a range of *m*/*z* = 70–1000, with the resolution set at 70,000, the AGC target at 1 × 10^6^, and the maximum injection time (Max IT) at 20 msec. For nucleotide tSIM scans, the resolution was set at 70,000, the AGC target was 1 × 10^5^, and the max IT was 100 msec.

#### 2.12.4. Data Analysis and Quantitation

Manual curation and integration of chromatographic peaks were performed with TraceFinder 4.1 (Thermo Fisher Scientific, Waltham, MA, USA) using a 5 ppm mass tolerance and referencing an in-house library of chemical standards. Data from TraceFinder were further consolidated and normalized with an in-house R script: (https://github.com/FrozenGas/KanarekLabTraceFinderRScripts/blob/main/MS_data_script_v2.4_20221018.R (accessed on 22 November 2022)). Briefly, this script performs normalization and quality control steps: (1) extracts and combines the peak areas from TraceFinder output .csvs; (2) calculates and normalizes to an averaged factor from all mean-centered chromatographic peak areas of isotopically labeled amino acids internal standards within each sample; (3) filters out low-quality metabolites based on user inputted cut-offs calculated from pool reinjections and pool dilutions; (4) calculates and normalizes for biological material amounts based on the total integrated peak area values of high-confidence metabolites. In this study, the linear correlation between the dilution factor and the peak area cut-offs is set to RSQ > 0.95 and the coefficient of variation (CV) < 30%. Finally, data were Log transformed and Pareto scaled within the MetaboAnalyst-based statistical analysis platform [23] to generate principal component analysis (PCA), partial least squares-discriminant analysis (PLSDA), and heatmaps. The relative abundance of each analyte per treatment was plotted and comparisons between each treatment alone and the DMSO control and each treatment + FA and the DMSO + FA control were performed using an Ordinary one-way ANOVA with Šídák’s correction for multiple comparisons.

### 2.13. STAT3 Luciferase Reporter Assay

The STAT3-dependent luciferase reporter assay was performed as previously reported [24]. One day prior to the assay, the U3A STAT3 luciferase reporter cell line [24] was seeded in a 96-well plate at 8000 cells per well. Cells were then incubated with compounds at the indicated concentration, and each well was treated with either media control or stimulated with 10 ng/mL oncostatin M (OSM) and incubated for six hours at 37 °C. Luciferase activity was quantified using the Bright-Glo Luciferase Assay system (Promega) and a Luminoskan Ascent Luminometer (Labsystems). Collected luminescence readings were normalized to DMSO-treated controls. Stimulation with (+OSM) gave luminescence values around 2.5 times that of the unstimulated control (−OSM). Quadruplicates were averaged and plotted with error bars representing standard deviation. Each condition was performed in quadruplicate on two separate occasions (*n* = 2) with representative data shown. Statistical significance was determined using Ordinary one-way ANOVA with a Dunnett multiple comparison test.

## 3. Results

### 3.1. Cyclogunail-Like Triazine Compounds Potently Inhibiti Human DHFR In Vitro

Given the high structural similarity between the antimalarials Cyc and Pyr (Figure 1a) and the recent evidence highlighting Pyr’s anti-cancer properties stem from functional inhibition of human DHFR [4], Cyc and its triazine analogues were analyzed to determine if they may interact with human DHFR using molecular docking. Cyc analogues from the NCI—Developmental Therapeutics Program were docked to human DHFR (PDB: 1U72) using GLIDE (Maestro, Schrödinger) with Xtra precision [16,17,25]. MTX was also docked into the active site as a positive control, which fit nearly identically compared to the co-crystalized MTX (RMSD: 0.964) (Figure 1b).

Docked Cyc analogues (57) were ranked according to their Xtra precision GLIDE (XPG) scores, and 40 compounds were acquired from the NCI. Compounds were selected based on their XPG score, their similarity to other compounds in the data set (prioritizing diversity), and based on their availability from the NCI. Notably, within these top compounds were several analogous pairs of compounds that possessed or lacked a chlorine atom on the phenyl substituent of the Cyc pharmacophore (i.e., 3,4-dichlorophenyl in 3077 *versus* 4-chlorophenyl in Cyc). Four compounds had an additional 4-*n*-butylphenyl substituent appended to the 4-position of the aromatic ring of the Cyc backbone, two of which also incorporated a sulfonylfluoride group that imparts the potential for covalently modifying DHFR (123032 and 127159). These compounds were initially reported as DHFR inhibitors back in the early 1970s [13,14,15]. Chlorinated analogues of Cyc (3077) and Pyr (3062), as well as Baker’s antifolate (139105), were included in this workflow. Baker’s antifolate has been well-characterized as a DHFR inhibitor and was previously a subject of Phase I clinical trials [26,27,28]. The structures of all compounds are shown in Figure 1a.

All compounds docked to the DHFR crystal structure with reasonably strong XPG scores (Table 1). Docking analyses predicted the phenyltriazine moiety of Cyc analogues (Figure 1c) and Cyc itself (Figure 1d) overlaid with the pteridine ring of MTX and the pyrimidine ring of Pyr within the DHFR active site. While hydrogen bonding with Ile7, Val115 and Glu30 anchor the nitrogen-containing heterocycles for MTX and Pyr, the triazine ring appears to rely more heavily on interactions with Glu30 making both ionic and hydrogen bonds with this residue. Cyc and Cyc analogues also made pi-stacking interactions with Phe34, and some made additional pi-stacking interactions with Phe31 (i.e., 128184 and 127153). Baker’s antifolate additionally made a hydrogen bond with Asn64. None of the Cyc compounds could mimic the ionic interactions MTX makes with Arg70 at the opening of the active site (Appendix A). Figure 1e shows the top docked pose of 127159 as a representative example highlighting the similarity in docked orientation relative to the Cyc backbone in Figure 1d. A notable difference between Cyc and 127159 docking orientations is that the triazine ring is flipped in how it aligns in the active site, which is most obvious when comparing the location of the dimethyl groups in Figure 1d,e. Ligand interaction diagrams for all docked compounds are included in Appendix A.

Encouraged by the docking scores and predicted interactions, these compounds were assessed for their ability to inhibit purified human DHFR enzymatic activity. DHFR activity was determined by monitoring reduced NADPH levels (absorbance at 340 nm) in the presence of dihydrofolic acid (DHF) and potential inhibitors. Percent inhibition was determined relative to the DMSO (no inhibition) and “no DHFR” (no activity) controls, and normalized data were fit to a four-parameter non-linear regression to determine IC_50_ values reported in Table 1. Inhibition curves are presented in Appendix A, and the linear range of the assay is presented in Appendix A.

In general, compounds possessing a 3-chloro moiety had lower IC_50_ values than analogous compounds, which lacked this group. This modification likely enhances the pi-stacking interaction with Phe34 seen in the docking simulations. The larger compounds 127159, 127153, and 128184 had IC_50_ values that were below 1 µM and approaching the IC_50_ of MTX (IC_50_ = 0.177 ± 0.006 μM). This correlated nicely with the docking scores, which suggested that these compounds fit well into the DHFR active site.

### 3.2. Cyclogunail-Like Triazene Compounds Elicit Anti-Cancer Activity in Breast Cancer Cells

All of the aforementioned compounds, except for 127153, were tested in the NCI-60 Human Tumor Cell Lines Screen to determine if they may have potential anti-cancer activity [19]. This publicly available data reports the concentration at which compounds induce 50% inhibition of cell growth (GI_50_) in an array of (male and female) cancer cell lines.

From the NCI-60 screens, all compounds displayed anti-cancer activity and were herein subjected to additional analyses to predict functional similarities between the compounds. To determine correlations, GI_50_ values for each compound against each cell line were compared, and Pearson correlation coefficients (r-values) were reported in Figure 2a. Data points where GI_50_ values were the same as the maximum concentrations tested were excluded from this analysis. While all compounds demonstrated a degree of positive correlation with the others, particularly strong correlations were observed within three different groupings: Firstly, MTX activity most strongly correlated with Pyr, the chlorinated Pyr analogue 3062 and Baker’s antifolate (139105), with r-values ranging from 0.67 to 0.74. Secondly, Cyc and its chlorinated analogue, 3077, unsurprisingly demonstrated a high degree of correlation (r = 0.76); however, these compounds did not correlate as strongly with the other analogues. In particular, Cyc demonstrated a relatively poor correlation with MTX (r = 0.40), suggesting that there may be differences in their functionality in these cell lines. Finally, the larger Cyc analogues (123032, 127159, 128184), as well as Baker’s antifolate (139105), had a high degree of correlation across the NCI-60 cell lines with r-values ranging between 0.73 and 0.91. The highest correlations were seen between chlorinated and non-chlorinated analogues (123032 and 127159, for example). All correlations were statistically significant, with *p*-values reported in Appendix A. These different groupings will be referred to herein as “MTX-like,” “Cyc-like,” or “Large Cyc Analogues,” where Baker’s antifolate spans both the “Large Cyc Analogues” and “MTX-like” groups.

To determine if these correlations were more pronounced in specific cancer types, Pearson correlation coefficients (r) were calculated within each cancer type, comparing compound GI_50_ values to MTX’s. Due to limited data availability, an r-value could not be determined for 3062 in leukemia. After plotting r-values as a heat map (Figure 2b), the strongest correlations are observed in melanoma and breast cancer cells, most of which were statistically significant (Appendix A). Further, GI_50_ values of each compound were plotted against breast cancer cell lines to investigate trends related to specific cell lines and potentially breast cancer subtypes (Figure 2c). These compounds typically displayed the strongest activity against MDA-MB-468 (triple negative) and MCF-7 (hormone receptor-positive, HER2 negative) and weakest activity against HS578T (triple negative), suggesting that their activity may not be linked to specific breast cancer subtypes. The Large Cyc Analogues appear to be more potent than other compounds in MDA-MB-231 cells. Based on these findings, MCF-7, MDA-MB-468, and MDA-MB-231 cells were used to investigate the effects of these compounds further.

### 3.3. Cycloguanil Analogues Inhibit Breast Cancer Cell Viability

Cell viability experiments in MDA-MB468, MDA-MB-231, and MCF-7 (female) breast cancer cell lines gave variable responses with the Cyc analogues despite the structural similarities and biochemical activity towards DHFR (Figure 3). In general, there are two distinct phenotypic responses detected in these cell lines. First, several compounds appear to induce a growth arrest, where viability remains around 50% relative to vehicle controls at high concentrations. These include Cyc-like and MTX-like compounds, including Baker’s antifolate. Second, the other Large Cyc Analogues killed cells at high concentrations and had minimal impact on viability at lower concentrations. Compounds 123023 and 127159 show little effect at low concentrations and cytotoxicity at higher concentrations in a dose-dependent manner. However, 127153 and 128184 induce growth arrest at low concentrations and induce cell death at higher concentrations. Thus, it appears there are two common phenotypic responses in these cell lines (growth arrest and inducing cell death), with Large Cyc Analogues behaving differently from the other compounds tested.

### 3.4. Folinic Acid Rescues DHFR Inhibitory Activity

In the folate metabolism cycle, DHFR reduces folic acid to dihyrdofolic acid and subsequently dihydrofolic acid to tetrahydrofolic acid, which is then further metabolized to produce metabolites needed for nucleic acid biosynthesis. Folinic acid (FA) is another dietary source for these nucleic acid building blocks, which does not require DHFR-mediated metabolism [29]. We previously observed that treatment with FA rescues DHFR-specific inhibitory mechanisms that impact downstream folate metabolism and cellular growth arrest and/or cell death [4]. To assess DHFR-dependence and FA rescue, cell viability assays were performed again in MDA-MB-468, MDA-MB-231, and MCF-7 cells in the presence of 10 μM of each compound ± 300 μg/mL FA for 72 h.

Consistently, in all three cell lines tested, FA rescued the effects of Pyr on cell viability. Only Pyr and Baker’s antifolate demonstrated statistically significant rescue in MDA-MB-468 cells, but viability impairment by Baker’s antifolate was not rescued in the other two cell lines (Figure 4 and Appendix A). Conversely, viability impairment by MTX was rescued in both MCF-7 and MDA-MB-231 cells but not in MDA-MB-468 cells. In general, Cyc-like compounds showed no rescue by FA in all cell lines, except for a small yet statistically significant rescue of 3077 viability impairment in MCF-7 cells. Interestingly, FA failed to rescue the impairment of viability caused by 127153 and 128184 in all three cell lines, whereas 127159 and 123032 were rescued in all but MDA-MB-468 cells, with the largest amplitude of rescue in MDA-MB-231 cells. The failure to rescue impaired viability with FA suggests that this effect may not be exclusively due to DHFR inhibition, and inhibition of other targets or pathways may contribute to the observed decreases in viability for some compounds.

### 3.5. Cycloguanil Derivatives Potently Engage DHFR in Cells and Cell Lysates

To determine whether each compound could engage DHFR in a biologically relevant environment, a modified cellular thermal shift assay (CETSA) in MDA-MB-468 lysates was performed. CETSA experiments measure changes in thermal stability that occur when a ligand or inhibitor binds to its protein target. A shift in the thermal denaturing profile of protein is evidence of direct physical interaction between an inhibitor and its target [30]. A melting curve for DHFR was established where cell lysates were first heated, then insoluble proteins were removed by centrifugation, and soluble DHFR levels were determined from the supernatant using Western blot. The quantified Western blot bands were fit using non-linear regression analysis to give an aggregation temperature (T_agg_, temperature where 50% of DHFR aggregates) of 41.50 ± 1.0 °C. When lysates were heated at 45 °C, there was ~18% soluble DHFR remaining (Appendix A), which was appropriate to investigate DHFR stabilization by inhibitors in a higher throughput experimental format. Uncropped blots are shown in Appendix A. MDA-MB-468 cell lysates were accordingly incubated with 10 μM of each inhibitor and heated at 45 °C for 3 min, which significantly reduced soluble DHFR levels compared to the 37 °C control (Figure 5a,b and Appendix A). Each compound substantially rescued DHFR against thermal degradation up to levels that were comparable to the DMSO control at 37 °C. For most treatments, this stabilization was statistically significant, except for 3062 and 139105 (Appendix A). Treatment with 10 μM Pyr, MTX, Cyc, 3077, 123032, and 127153 resulted in band densities that were greater than 100% which may be due to some denaturing occurring in the 37 °C control that the inhibitors can also rescue. Uncropped blots are shown in Appendix A.

To move into a cellular context, DHFR accumulation was used as a determinant of target engagement leveraging DHFR’s autoregulatory *m*RNA-binding mechanism. In cells, human DHFR binds to its own *m*RNA, preventing further translation into protein [31]. However, upon inhibitor binding to DHFR, this interaction is disrupted, releasing DHFR *m*RNA, which is translated into protein. This leads to a temporary accumulation of DHFR protein levels which can be observed in both cell-based assays [4] and clinical settings [32,33].

All compounds tested led to an accumulation of DHFR in MDA-MB-468 breast cancer cells in a dose-dependent manner, indicating that they engaged DHFR in cells (Figure 5c). Pyr and MTX both induced accumulation of DHFR in a similar manner to previous studies [4]. Several of the Cyc derivatives were more potent than both Pyr and MTX, approaching saturating DHFR accumulation at a concentration of only 1 nM. Subsequent experiments were performed at a lower concentration range (0.012–1 nM) where MTX, 127153, 127159, and 128184 led to a notable accumulation of DHFR at only 0.33 nM (Figure 5d), indicating these triazene compounds are highly potent and cell-active DHFR inhibitors. Uncropped Western blots from all DHFR accumulation experiments are shown in Appendix A.

### 3.6. Inhibitors Fail to Impair Wound Healing nor bind Thymidine Phosphorylase in MDA-MB-231 Breast Cancer Cells

A recent report highlighted Pyr demonstrates promising anti-cancer and anti-metastatic activity in lung cancer through dual targeting of DHFR and thymidine phosphorylase (TP) [34]. To determine if targeting TP may also contribute to the reduced viability of breast cancer cells, TP engagement by CETSA was assessed MDA-MB-468 cell lysates, but no stabilization of TP was detected by any of the compounds, including Pyr (Appendix A).

To investigate the effects on cell migration, a wound-healing assay was performed in MDA-MB-231 cells. A scratch wound was induced on a confluent layer of MDA-MB-468 cells treated with or without 10 μM inhibitor. In this setting, none of the inhibitors significantly inhibited cell migration relative to the DMSO control (Appendix A). Taken together, these data failed to confirm Pyr and other DHFR inhibitors bind to TP or inhibit migration of MDA-MB-231 cells at concentrations where Pyr, MTX, Cyc, and Cyc derivatives engage DHFR.

### 3.7. Cycloguanil and 127159 Inhibit Cellular Folate Metabolism

Out of the Cyc analogues tested, 127159 was selected for further study due to its robust interaction with DHFR in vitro and cells, as well as strong rescue by FA in cell viability experiments, especially in MDA-MB-231 cells. Therefore, 127159 and Cyc were assessed alongside MTX and Pyr (tested previously [4]) to determine their impact on folate-dependent nucleotide levels. Accordingly, MDA-MB-231 cells were treated with MTX, Pyr, Cyc, or 127159 (10 μM) with or without 300 μg/mL FA, and nucleotide levels were assessed using LC/MS. Supplementation with FA resulted in an increase in downstream folate species, as expected (Appendix A). Principal component analysis (PCA) of the metabolic response to the different compounds (with and without FA supplementation) demonstrated that Cyc- and MTX-treated cells clustered together, whereas 127159 clustered with Pyr (Figure 6a). All four treatments were markedly different compared to the DMSO-treated cells. Cells treated with MTX, Pyr, and Cyc and supplemented with FA clustered with DMSO-treated cells, whereas the 127159 + FA trended towards these data but remained distinct (Figure 6a). This suggests a common molecular mechanism for the four drugs involved in reduced-folate depletion via DHFR inhibition.

To investigate the downstream impacts of these compounds, nucleotide metabolite levels were assessed (summarized in Figure 6b). All four compounds significantly depleted ADP, CDP, GDP, and IMP (Figure 6c and Appendix A). UDP was significantly depleted in cells treated with Pyr and Cyc but not upon treatment with MTX and 127159 alone. In all cases, FA supplementation led to an accumulation of nucleotides approaching the DMSO + FA control, although not all conditions were statistically insignificant compared to the control (Appendix A). Other nucleotide intermediates, including those which rely on reduced folate for their metabolism (dUMP, glycineamide ribonucleotide (GAR), 5-aminoimidazole-4-carboxamide ribonucleotide (AICAR)) and the pyrimidine intermediate orotate were also measured. Similar effects were observed on dUMP and GAR, where DHFR inhibitor treatment increased these nucleotides, then returned them towards control levels with FA, although 127159 + FA quantitation of GAR was significantly different from the DMSO + FA control (Figure 6d and Appendix A).

Interestingly, 127159 had strikingly different impacts on orotate and AICAR levels compared to the other anti-folates. While decreased or statistically insignificant changes in orotate levels were observed in cells treated with MTX, Pyr, or Cyc, treatment with 127159 led to a stark increase in orotate levels. FA supplementation returned orotate to vehicle control levels for MTX, Cyc, and Pyr; however, it failed with 127159, where orotate remained elevated in the presence of FA. AICAR levels also increased slightly, yet significantly after treatment with 127159, whereas Cyc and Pyr induced much larger increases in AICAR and MTX failed to increase AICAR levels in a statistically significant manner. Treatment with FA returned AICAR to the same levels as vehicle controls for Cyc and Pyr, but FA treatment further increased AICAR by 127159 (Figure 6c,d), which may suggest that 127159 inhibits downstream processing of AICAR. Together, these results corroborate the hypothesis that Cyc and 127159, such as MTX and Pyr, likely inhibit DHFR to impact cellular metabolite levels, although there are some unique aspects of their impact on nucleotide metabolism that may warrant further study.

### 3.8. Cycloguanil Analogues Inhibit STAT3 Transcriptional Activity

Finally, with recent evidence demonstrating that Pyr and MTX induce their anti-cancer activity (at least in part) by halting STAT3 transcriptional activity, these compounds were assessed in the STAT3-dependent luciferase reporter assay using U3A (male) fibrosarcoma cells, as performed previously [4]. This cell line lacks basal STAT3 phosphorylation and subsequent activation [4] and lacks expression of STAT1 [35], thus eliminating confounding STAT1-STAT3 interactions and background STAT3-dependent signal. A broad concentration range (0.02–20 μM) was used to cover both the concentrations where DHFR accumulation was observed and where phenotypic responses were seen. Interestingly, potent inhibition of luciferase activity is found with MTX and Cyc even at low inhibitor concentrations, whereas Pyr and 127159 demonstrate a dose-response and do not reach full STAT3 inhibition until µM concentrations (Figure 7). Data from an additional independent experiment are shown in Appendix A which shows a similar trend and statistical comparisons for these experiments are displayed in Appendix A. Combined, these results support previous claims that targeting DHFR decreases STAT3 transcriptional activity and point towards STAT3 being involved in the anti-cancer mechanism of these DHFR inhibitors.

## 4. Discussion

Current anti-cancer drug treatment strategies include diverse medicines spanning more than 70 years of anti-cancer drug discovery research—from state-of-the-art targeted therapies and biologics all the way back to early anti-metabolites and platinum-based chemotherapies. While several classic therapies have stood the test of time and remain the standard of care for cancer patients, the search for new and more effective cancer therapies remains an eternal quest for drug discovery scientists. In this study, modern chemical biology techniques and analytical methods have been applied to classic DHFR inhibitors MTX, Pyr, Cyc, and Cyc analogues. Capitalizing on databases from the NCI—Developmental Therapeutics Program and NCI-60 Human Tumor Cell Lines Screening, compounds with promising inhibitory activity against human DHFR were revisited [16,19]. These Cyc analogues were initially explored for their anti-cancer activity and ability to inhibit DHFR in tissue isolates and in vivo models [14,36]. While Cyc, Pyr, MTX, and Baker’s antifolate had a considerable mention in the literature, the other Large Cyc Analogues appear in relatively few resources. In early studies in the 1970s, 123032 (compound 10 in reference [14]), 128184 (compound 24 in reference [15]), 127153 (compound 19 in reference [14]), and 127159 (compound 2 in reference [13]) showed promising anti-leukemic activity in vivo, even compared to MTX. Thus, their emergence in this study, using in silico docking and historical growth inhibition from the NCI, was somewhat surprising, yet highlights the utility of drug repurposing and revisiting even very early anti-cancer compounds to define their scope of use better and to explore them as potential chemical probes.

The NCI-60 data indicated a high correlation between inhibitor GI_50_ values in both breast cancer and melanoma cell lines. While this study focused on breast cancer, further studying these compounds within the context of melanoma is recommended. In-house generated cell viability data reinforced trends observed in the NCI data sets and previous studies with these compounds [13,14,15,36,37]. Many of the compounds gave dose-response curves that had lower asymptotes, around 50% viability, suggesting a growth inhibition rather than a cytotoxic mechanism of action. In addition to the growth inhibition phenotype, some compounds also induced cell death at higher concentrations. Initially, this was suspected to be evidence of polypharmacology or non-specific toxic effects of compounds; however, FA rescue experiments did not support this hypothesis, where impaired viability of some of the cell death-inducing compounds (i.e., 123032 and 127159) could be rescued by FA in MCF-7 and MDA-MB-231 cells. Intriguingly, these compounds both possess a sulfonyl fluoride moiety which confers covalent reactivity to these inhibitors and can be associated with promiscuity. Their rescue by FA, and the lack of rescue with the non-sulfonyl fluoride analogous compounds (127153 and 128184), in fact, suggests that the sulfonyl fluoride moiety may instead confer selectivity for DHFR rather than promiscuity.

Although a previous study suggested that Pyr is also a direct binder and inhibitor of TP [34] in biochemical experiments (Biacore and differential scanning fluorimetry), CETSA experiments with Pyr, MTX, Cyc, and Cyc analogues could not confirm binding to TP in breast cancer cell lysates at relevant concentrations for DHFR engagement and phenotypic responses in breast cancer cell lines. These compounds also failed to halt cell migration in the wound-healing assay. Similarly, MTX has recently been shown not to impact migration in MDA-MB-468 cells [38] but has been found to promote migration in other cancer cell types [39]. The data presented herein, along with recent investigations on the impact of nucleic acid synthesis, may point toward cell line-specific or cancer-type-specific responses to these inhibitors.

Perhaps most striking of all is the DHFR target engagement by accumulation data, where Cyc analogues were extremely potent. Similar to MTX, several of the Cyc compounds showed evidence of DHFR binding below 1 nM concentration, whereas Pyr and Cyc, required higher concentrations to induce DHFR accumulation. Some Cyc analogues induced comparable DHFR accumulation to MTX, where 10 nM induced a ~5-fold increase, and 1 µM induced ~40-fold increase in DHFR levels, compared to 127159, which induced 10- and 30-fold increases at 10 nM and 1 µM, respectively. Moreover, while Cyc failed to induce DHFR accumulation at low nM concentrations in MDA-MB-468 cells, STAT3-dependent luciferase activity was significantly inhibited upon treatment with 20 nM Cyc in U3A. This may arise either due to cell-line differences or may support a different target or mechanism that may be linked to the STAT3 inhibitory activity of these compounds.

Finally, the metabolite profiles for these compounds were remarkably similar when analyzed by PCA, where Cyc and Pyr clustered together, while MTX and 127159 clustered together, all of which moved closer to DMSO-treated levels upon treatment with FA. However, deeper analyses identified unique ways of 127159 affecting the metabolite pools, namely by increasing AICAR levels in the presence of FA, whilst other DHFR inhibitors tended to give the opposite response. In addition, 127159 increased orotate levels both with and without FA supplementation, which was unique to this compound over other DHFR inhibitors tested. Orotate is a pyrimidine molecule that can be converted into uridine monophosphate by uridine-5-monophosphate synthetase. The buildup of orotate by 127159 uniquely is curious and points to possible off-target effects of the inhibitor, but further investigation is required to investigate this phenomenon. These effects are likely the underlying reasons for the differences in the PCA for this compound relative to the other DHFR inhibitors.

Overall, it is reasonable to suspect that these differences may be of biological significance and linked to the anti-cancer efficacy of these compounds; however, at this time, the precise nature of this relationship is not yet clearly defined for all of these compounds and warrants further study.

## 5. Conclusions

The presented data highlight the benefits of applying state-of-the-art chemical biology and analytical techniques to classic small molecule inhibitors. These methods have shown 127159 and other Cyc analogues behave as potent DHFR inhibitors and may be considered for further exploration within the realm of cancer therapies, perhaps with greater emphasis on specific disease models and/or scope of utility.

## Figures and Tables

**Figure 1 metabolites-13-00151-f001:**
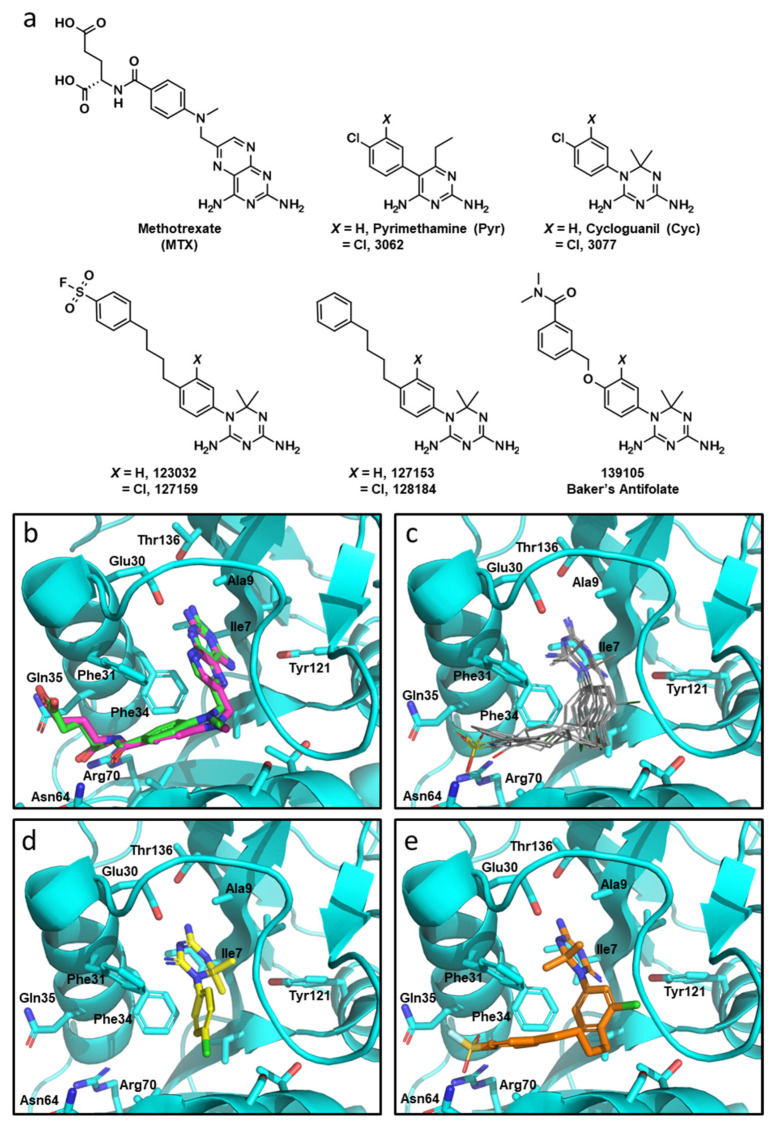
Molecular docking with Cyc and Cyc analogues to human DHFR. (**a**) Chemical structures of DHFR inhibitors methotrexate (MTX), Pyrimethamine (Pyr), Cycloguanil (Cyc), and Baker’s antifolate, as well as analogous top compounds from the NCI-DTP database referred to by their respective NSC numbers. (**b**) Overlay co-crystalized MTX (green) and docked MTX (magenta) into crystalized DHFR (PDB: 1U72) using GLIDE (Maestro, Schrödinger). NADPH was removed from the active site prior to the docking simulation. (**c**) Overlaid docking images of Pyr, 3062, Cyc, 3077, 123032, 127159, 127153, 128184, and 139105 in the folate-binding pocket of DHFR. The diamino nitrogen-containing heterocycles of these compounds mimic the interactions of the diaminopteridine moiety of MTX. (**d**) The top scoring pose of Cyc (yellow) and (**e**) 127159 (orange) docked into the folate-binding pocket of DHFR.

**Figure 2 metabolites-13-00151-f002:**
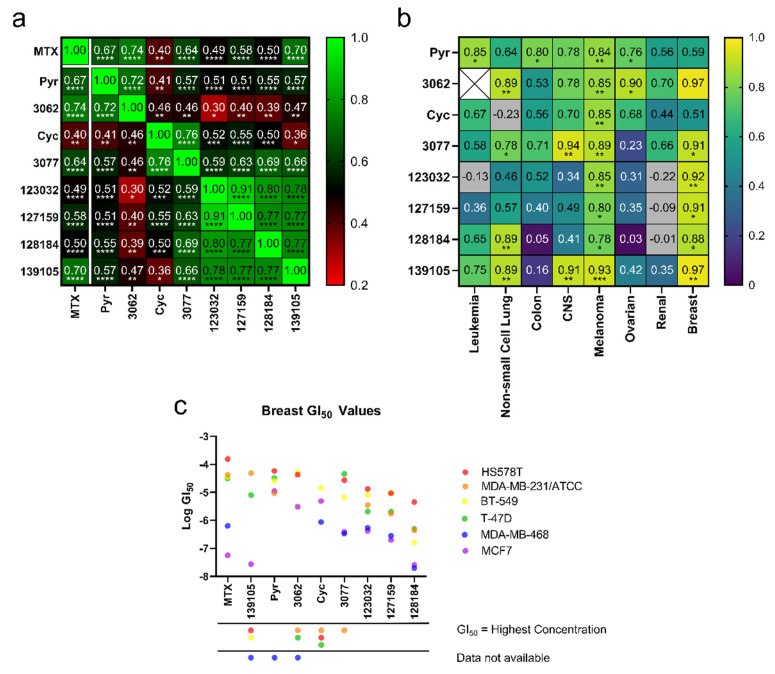
Cyc analogues inhibit cell growth in cancer cells. (**a**) GI_50_ values obtained from the NCI-60 Human Tumor Cell Lines Screen were compared against each compound to reveal statistically significant Pearson Coefficient correlations. (**b**) GI_50_ values of each compound compared to MTX separated by cancer type reveal positive, statistically significant correlations of Pearson Coefficients in melanoma and breast cancer cells. There were not enough data to compare 3062 to MTX in leukemia (white X box). Grey boxes indicate a negative correlation outside the heatmap range. The statistical significance of both plots in (**a**,**b**) was determined using a two-tailed statistical analysis. (**c**) Plots of the GI_50_ values for each compound against each breast cancer cell line. For instances where GI_50_ values were equal to the maximum concentration tested, data were omitted, as indicated by the symbols under the x-axis. (ns = not significant; * = *p* ≤ 0.05; ** = *p* ≤ 0.01; *** = *p* ≤ 0.001; **** = *p* ≤ 0.0001).

**Figure 3 metabolites-13-00151-f003:**
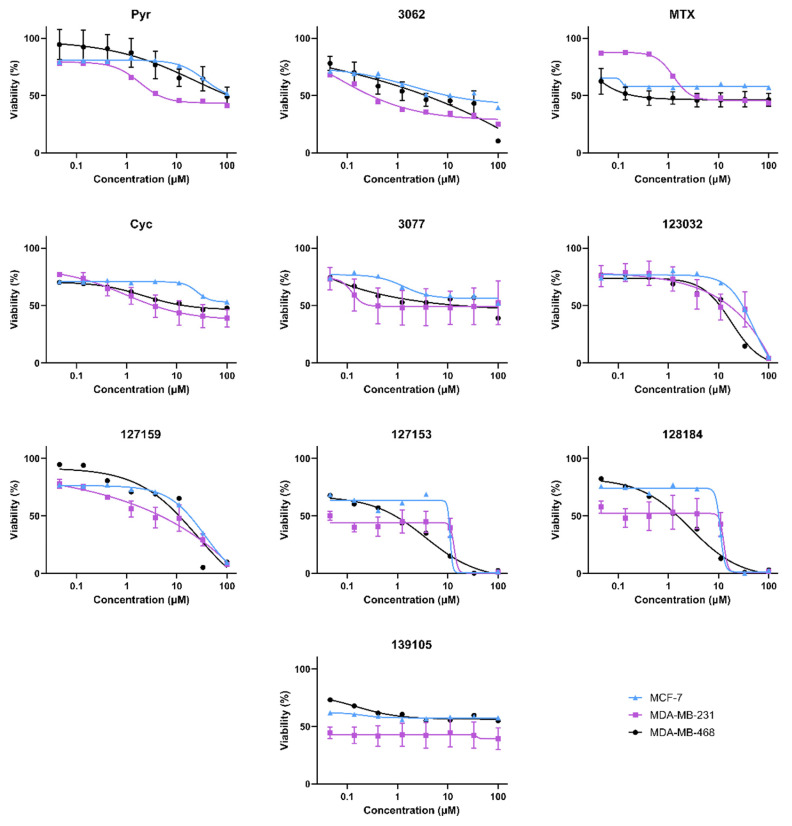
Biological activity of Cyc analogues in breast cancer cells. Cell viability of MDA-MB-468 (black circle), MDA-MB-231 (purple square), and MCF-7 (blue triangle) breast cancer cells when treated with methotrexate and cycloguanil analogues. Cells were treated with 0–100 μM of each inhibitor for 72 h. Cell viability was measured by adding 44 μM resazurin and quantifying the amount brought into and metabolized by live cells by monitoring the fluorescence of Ex_540_/Em_600_. Percent viability relative to the DMSO control was calculated and plotted against the inhibitor (μM, Log_10_ scale). Models of the data were determined by fitting Equation (2).

**Figure 4 metabolites-13-00151-f004:**
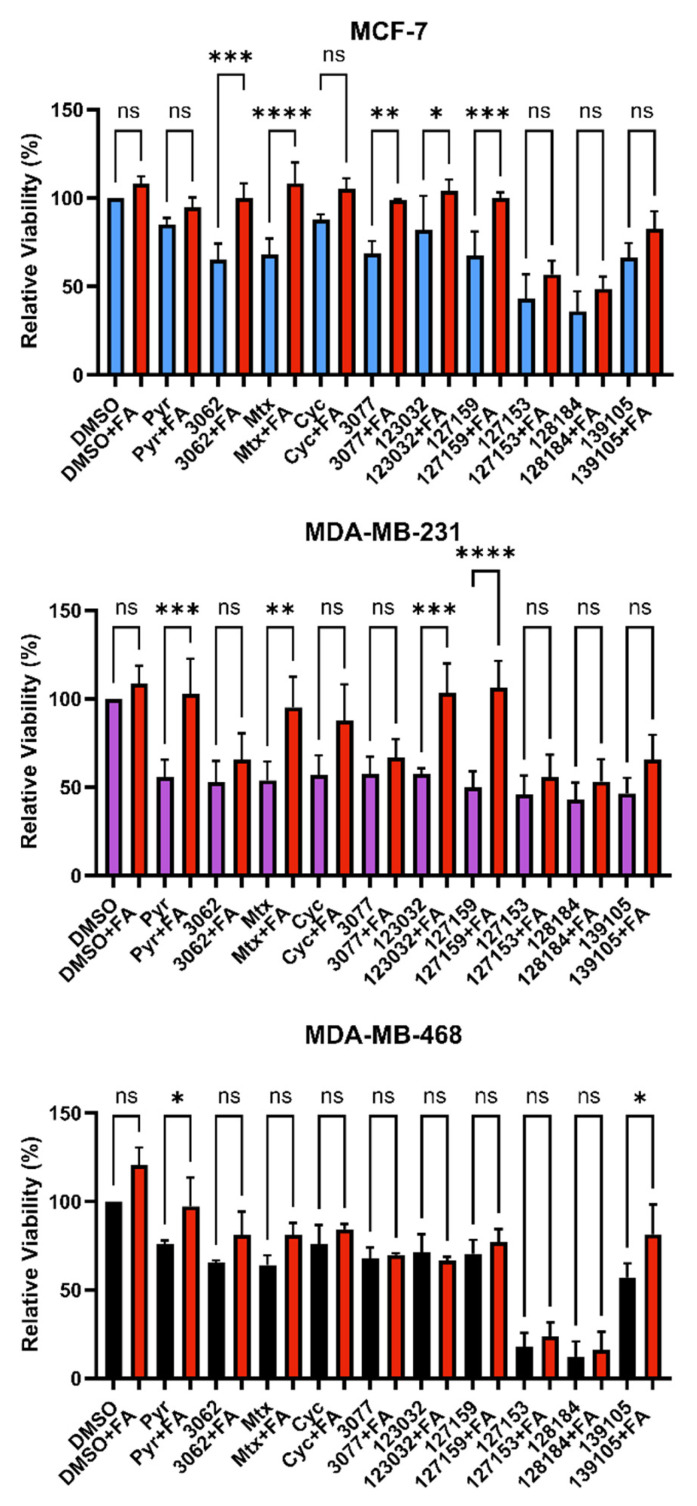
Folinic acid viability rescue in breast cancer cells. MCF-7, MDA-MB-231, and MDA-MB-468 cells were treated with 10 μM inhibitor for 72 h with or without 300 μg/mL folinic acid (FA) to investigate DHFR dependence. Cell viability was monitored by the incorporation of resazurin, as measured by fluorescence. The relative viability of each treatment was normalized to the DMSO control. An Ordinary one-way ANOVA with Šídák’s multiple comparisons was performed to test for significant differences between each treatment with inhibitor alone and treatment with inhibitor and FA (red). Each experiment was performed in triplicate and repeated in three independent assays (*n* = 3). (ns = not significant; * = *p* ≤ 0.05; ** = *p* ≤ 0.01; *** = *p* ≤ 0.001; **** = *p* ≤ 0.0001).

**Figure 5 metabolites-13-00151-f005:**
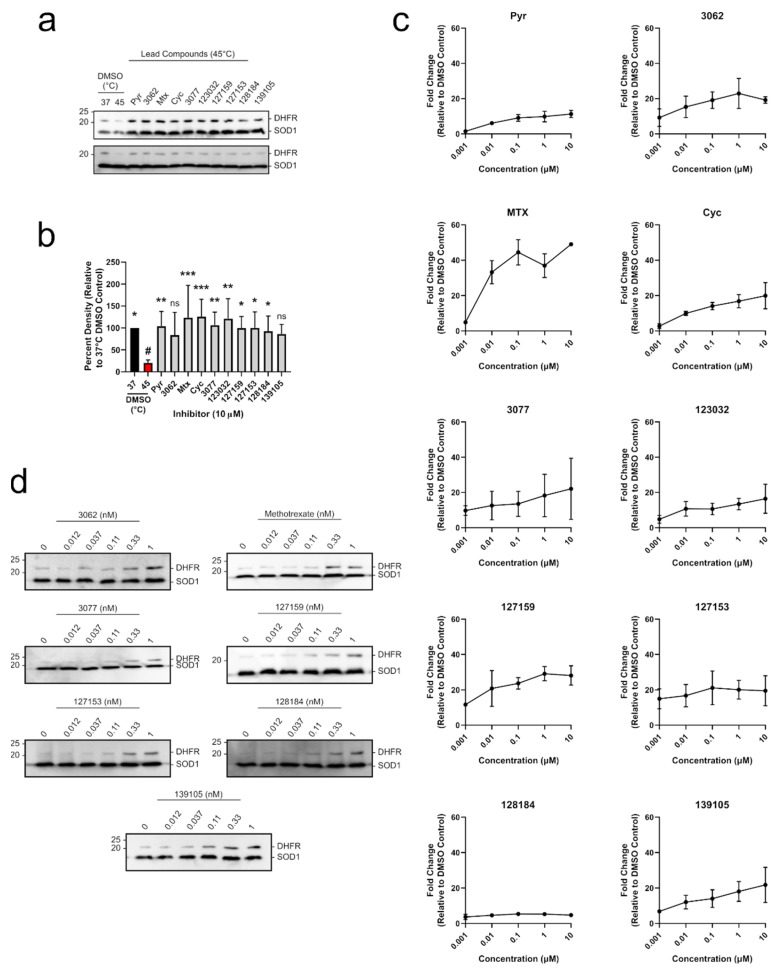
Cyc analogues interact with and stabilize DHFR in breast cancer cells. (**a**) MDA-MB-468 lysates were treated with 10 μM of each compound and incubated at 45 °C. Lysates were electrophoretically separated, and DHFR was detected via Western blot. DHFR bands were normalized to the thermostable SOD1 control. (**b**) Relative band intensities from (**a**) were compared to the DMSO control at 45 °C (red, #) using an Ordinary one-way ANOVA with Dunnett’s test for multiple comparisons. Each treatment was performed with five replicates over three independent experiments (*n* = 5). (**c**) MDA-MB-468 cells were treated with 0–10 μM of each inhibitor for 24 h. DHFR and the thermostable SOD1 were detected via Western blot as described in (**a**). Relative band densities were plotted against inhibitor concentration (μM, Log_10_ scale) to observe a dose-dependent increase in DHFR protein levels. (**d**) MDA-MB-468 cells were treated with 0–1 nM of select inhibitors as described in (**c**) to investigate relative potency. Some compounds stabilize DHFR at sub-nanomolar concentrations. All DHFR accumulation experiments were performed in duplicate in two independent assays (*n* = 2). (ns = not significant; * = *p* ≤0.05; ** = *p* ≤ 0.01; *** = *p* ≤ 0.001).

**Figure 6 metabolites-13-00151-f006:**
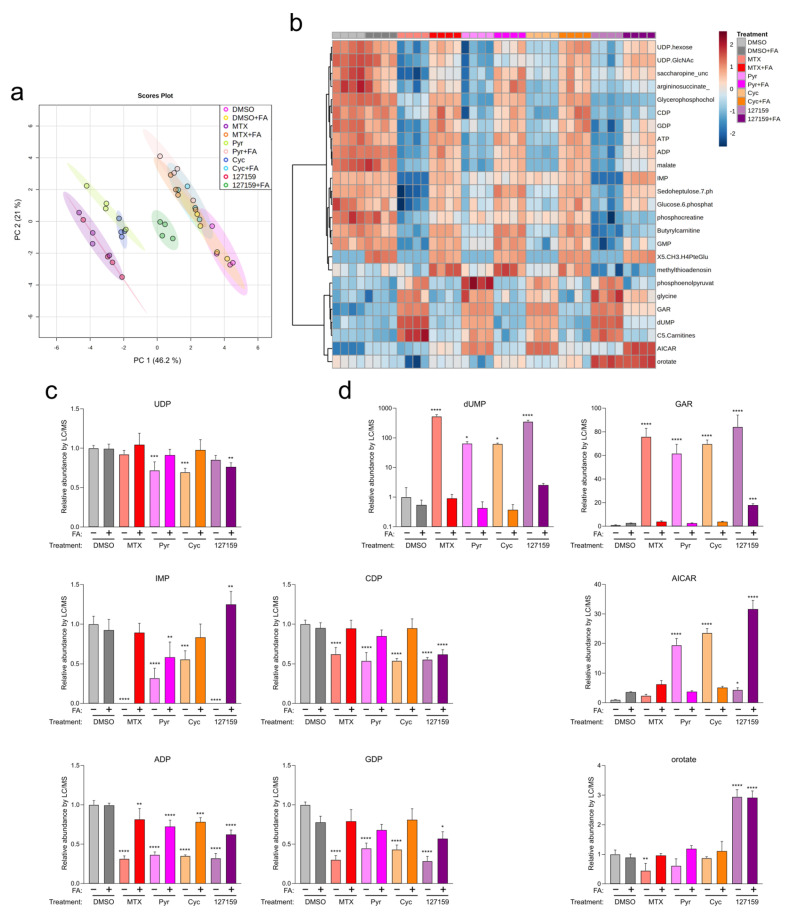
Metabolite profiling with FA supplementation support Cyc and 127159 impact folate homeostasis through inhibition of DHFR. (**a**) Global PCA analysis of polar metabolites detection in cells treated for 24 h with DMSO, MTX, Pyr, Cyc, 127159, and ± folinic acid. (**b**) Top 25 differentially detected metabolites from (**a**). (**c**) Detected levels of representative nucleotides of de novo synthesis and salvage pathways in the treated cells from (**a**). All levels were normalized to the DMSO level. (**d**) Detected levels of nucleotide intermediates in the treated cells from (**a**). Each replicate represents extracted metabolites from one million cells per condition, and error bars represent the standard deviation of the mean (*n* = 4). All levels were normalized to the DMSO level. An Ordinary one-way ANOVA with Šídák’s correction for multiple comparisons was performed to compare each treatment versus DMSO control and to compare each treatment + FA versus DMSO + FA control. Only statistically significant comparisons are indicated (* = *p* ≤0.05; ** = *p* ≤ 0.01; *** = *p* ≤ 0.001; **** = *p* ≤ 0.0001).

**Figure 7 metabolites-13-00151-f007:**
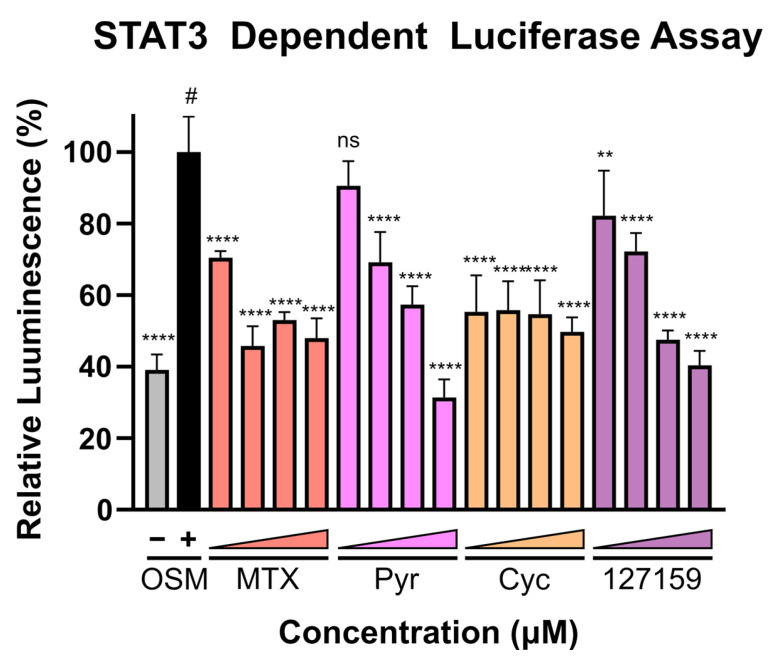
DHFR inhibitors block STAT3-dependent gene expression. U3A cells were incubated with respective inhibitors at 0.02, 0.2, 2, or 20 μM for 1 h prior to the addition of OSM (10 ng/mL) or vehicle control and incubation for 5 h. Luciferin was added, and luminescence was measured. Presented data are normalized to the OSM stimulated control (100%, #), and statistical significance was determined using an Ordinary one-way ANOVA with a Dunnett multiple comparison test (ns = not significant; ** = *p* ≤ 0.01; **** = *p* ≤ 0.0001). This is a representative figure of the experimental results. Data from an additional individual experiment are shown in Appendix A.

**Table 1 metabolites-13-00151-t001:** Comparison of GLIDE XPG scores and biochemical IC_50_ values in the DHFR activity assay. Errors represent the standard deviation between two independent experiments (*n* = 2), each performed in duplicate.

Compound	XPG Score (kcal/mol)	IC_50_ (μM)
MTX	−12.630	0.117 ± 0.006 ^a^
Pyr	−8.217	16.9 ± 7.9
3062	−8.774	8.6 ± 2.7
Cyc	−7.718	10.8 ± 3.5
3077	−8.653	1.26 ± 0.66
123032	−10.067	1.17 ± 0.46
127159	−10.257	0.75 ± 0.32
127153	−9.338	0.93 ± 0.24
128184	−10.193	0.72 ± 0.24
139105	−10.165	2.11 ± 0.53

^a^ average of three replicates reported for MTX in Heppler et al., 2022 [4].

## Data Availability

All data from the National Cancer Institute Developmental Therapeutics Program NCI-60 Human Tumor Cell Lines Screen can be found here: https://dtp.cancer.gov/public_compare/. All data generated in this study are available upon request from the corresponding author.

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
