# Peer review of "Cycloguanil and Analogues Potently Target DHFR in Cancer Cells to Elicit Anti-Cancer Activity"

_metabolites, 2023, doi:10.3390/metabo13020151_

Round 1

Reviewer 1 Report

In the manuscript by Brown et al, the authors re-investigate effects of Cycloguanil and related compounds on DHFR by applying computational and biochemical assays. They compare the results to the classical DHFR inhibitor Methotrexate. The compounds are studied in three breast cancer cell lines MDA-MB-231, MDA-MB-468 and MCF7. Rescue experiments are performed with folinic acid. The authors also report that the compounds have no effect on cell migration. This phenomenon has recently been studied in detail by two independent publications which the authors may want to take into consideration (doi: 10.1038/s41467-022-30362-z and 10.1038/s41467-022-30363-y). Finally, they assess STAT3 activity by Luciferase reporter assay in U3A fibrosarcoma cells. This final experiment is not ideal as it is in a completely different cell model where effects on DHFR are not presented. However, given the direct comparison to MTX the data is still of interest. Overall, the authors confirm that these compounds have inhibitory effects on DHFR and uncover some distinct (and interesting) differences that can be followed up in future research.

In sum, I have no major concerns and propose that the manuscript can be published in its present form. I would only include the µM numbers in x-axis of figure 7 (or provide them in the figure legend). In the current version it only indicates “concentration (µM)” without providing the actual numbers.

Reviewer 2 Report

The manuscript revisits known DHFR inhibitors and applied newer chemical biology, analytical and computational techniques in an attempt to verify the reported data and elucidate the mechanism of action of these compounds. Overall, the manuscript is well written with significant characterization of the NCI leads. The reviewer has a few comments that need to be addressed before the manuscript can be published in the journal.

Figure 1a should be mentioned in the introduction, especially when discussing the structural similarity between the compounds.

The authors mention that “Despite showing promising activity in these early experiments, some of these compounds have seemingly not been further explored as potential cancer therapeutic agents.” Discussion should be added on why this is the case. Additionally, literature is not cited for this statement.

In section 3.1. the authors should mention details on the number of cyc analogs that were initially docked, what XPG score cut-off was set to deem a compound a hit for purchasing, and how many compounds were finally purchased.

Although Figure 1b represented the overlap between the docked and co-crystalized poses of MTX, RMSD between these should be mentioned as a validation of their docking protocol.
